# In Silico and In Vitro Studies of Benzothiazole-Isothioureas Derivatives as a Multitarget Compound for Alzheimer’s Disease

**DOI:** 10.3390/ijms232112945

**Published:** 2022-10-26

**Authors:** Martha Cecilia Rosales Hernández, Leticia Guadalupe Fragoso Morales, José Correa Basurto, Marycruz Olvera Valdez, Efrén Venancio García Báez, Dania Guadalupe Román Vázquez, Ana Paola Anaya García, Alejandro Cruz

**Affiliations:** 1Laboratorio de Biofísica y Biocatálisis, Sección de Estudios de Posgrado e Investigación, Escuela Superior de Medicina, Instituto Politécnico Nacional, Plan de San Luis y Díaz Mirón s/n, Ciudad de Mexico 11340, Mexico; 2Laboratorio de Diseño y Desarrollo de Nuevos Fármacos e Innovación Biotecnológica, Sección de Estudios de Posgrado e Investigación, Escuela Superior de Medicina, Instituto Politécnico Nacional, Plan de San Luis y Díaz Mirón, Ciudad de Mexico 11340, Mexico; 3Laboratorio de Nanomateriales Sustentables, Sección de Estudios de Posgrado e Investigación, Escuela Superior de Ingeniería Química e Industrias Extractivas, Instituto Politécnico Nacional, Av. Instituto Politécnico Nacional, s/n, Unidad Profesional Adolfo López Mateos, Ciudad de Mexico 07708, Mexico; 4Laboratorio de Investigación en Química Orgánica y Supramolecular, Unidad Profesional Interdisciplinaria de Biotecnología del Instituto Politécnico Nacional, Av. Acueducto s/n Barrio la Laguna Ticomán, Ciudad de Mexico 07340, Mexico

**Keywords:** benzothiazoles, isothioureas, in silico, multitarget, amyloid beta, AChE

## Abstract

Alzheimer’s disease (AD) is a progressive neurodegenerative disorder. Inhibiting acetylcholinesterase (AChE), amyloid beta (Aβ_1-42_) aggregation and avoiding the oxidative stress could prevent the progression of AD. Benzothiazole groups have shown neuroprotective activity whereas isothioureas groups act as AChE inhibitors and antioxidants. Therefore, 22 benzothiazole-isothiourea derivatives (**3a**–**v**) were evaluated by docking simulations as inhibitors of AChE and Aβ_1-42_ aggregation. In silico studies showed that **3f**, **3r** and **3t** had a delta G (ΔG) value better than curcumin and galantamine on Aβ_1-42_ and AChE, respectively. The physicochemical and pharmacokinetics predictions showed that only **3t** does not violate Lipinski’s rule of five, though it has moderated cytotoxicity activity. Then, **3f**, **3r** and **3t** were synthetized and chemically characterized for their in vitro evaluation including their antioxidant activity and their cytotoxicity in PC12 cells. **3r** was able to inhibit AChE, avoid Aβ_1-42_ aggregation and exhibit antioxidant activity; nevertheless, it showed cytotoxic against PC12 cells. Compound **3t** showed the best anti-Aβ_1-42_ aggregation and inhibitory AChE activity and, despite that predictor, showed that it could be cytotoxic; in vitro with PC12 cell was negative. Therefore, **3t** could be employed as a scaffold to develop new molecules with multitarget activity for AD and, due to physicochemical and pharmacokinetics predictions, it could be administered in vivo using liposomes due to is not able to cross the BBB.

## 1. Introduction

Alzheimer’s disease (AD) is a neurodegenerative disorder that causes cognitive impairment mediated by the senile plaques and neurofibrillary tangles formation in the brain [1]. Currently, AD is the leading cause of dementia (60–70%) among older adults worldwide, being the sixth-leading cause of death in the United States [2]. Despite the efforts trying to elucidate the exact etiology of AD, it has not been possible yet since the final diagnosis can only be made after death. This is due to the complex neuropathology of AD associated with several risk factors such as: aging, previous head injuries, vascular risk factors from diabetes, hypercholesterolemia and hypertension [3,4].

Then, the complexity of the AD physiopathology, together with various pathologies already in AD patients, suggests that traditional drugs are not viable for adequate therapeutic effect [5,6]. In this sense, recent reports have been focused on evaluated multitarget compounds as possible treatments for AD as a more appropriate approach [7,8]. An interesting bibliometric (1990–2020) was published recently describing the biological targets and mechanism of multitarget anti-Alzheimer’s drugs, the most important being Acetylcholinesterase (AChE), Butirilcholinesterase (BChE), monoamine oxidase A (MAO A) and monoamine oxidase B (MAO B), beta-secretase 1 (BACE1), oxidative stress (OS), biometals and amyloid beta (Aβ_1-42_) aggregation, of which a binary combination is the most employed [9]. Many multitarget compounds have been designed to inhibit AChE and another biological target using the chemical core of AChE inhibitors. Furthermore, a multitarget compound for AD has been identified by virtual screening from traditional Chinese medicine [10]. 

Therefore, the design of multitarget compounds based on AD physiopathology involved compounds that inhibit AChE, an enzyme responsible to hydrolyze the acetylcholine neurotransmitter (ACh) and related to cholinergic hypothesis. This justifies the use of AChE inhibitors (AChEI) such as galantamine for AD [11]. Another enzyme is the beta-secretase 1 (BACE1), which is a transmembrane aspartic protease, which cleaves amyloid precursor protein (APP) and produces Aβ_1-42_ [12,13]. The Aβ_1-42_ can aggregate and produce neurotoxicity, thus, inhibiting BACE1 can help in the treatment of AD [14,15]. In this context, in 2020 the Food and Drug Administration (FDA) approved the use of an antibody (Aduhelm from Biogen) to reduce the amyloid beta plaques in the brain as a treatment for AD [16,17].

Taking into account that benzothiazole is a heterocycle found in a variety of pharmaceutical drugs, this pharmacophore could be useful in AD treatment [18,19]. For instance, riluzole (2-amino-6-trifluoromethoxy) benzothiazole is used to treat amyotrophic lateral sclerosis due to its neuroprotective effect acting as voltage-gated sodium channel blockers, noncompetitive inhibition of NMDA receptors and inhibition of glutamate release [20]. Recently, an evaluation of riluzole in a clinical trial for AD was reported [21]; the results showed that riluzole affects the glucose metabolism and glutamate levels. However, this study did not show effects on Aβ [20]. Riluzole was also evaluated employing ABPP/PS1 mice (a cerebral amyloidosis model), showing that glutamate concentrations were maintained in both the control mice and the ABPP/S1 group receiving riluzole. In addition, the treatment of ABPP/PS1 mice with riluzole prevented their cognitive decline. However, ABPP/PS1 mice under treatment with riluzole did not showed effects on Aβ accumulation [22].

Therefore, results interesting to evaluate new molecules that combine the benzothiazole and isothiourea groups and added other chemical substituents to develop new molecules to prevent both the Aβ aggregation and AChE activity and with antioxidant activity. Some isothioureas has antioxidant activity and exhibits inhibitory activity against AChE [23]. 

Consequently, in this work we proposed to evaluate 22 benzothiazole-isothiourea derivatives (Figure 1) by docking simulations. We selected those capable of making chemical interactions with amino acids residues of the AChE active site and those involved in the Aβ_1-42_ aggregation. Considering the binding mode and the free energy (ΔG) values from in silico studies, the best compounds were selected; **3f**, **3r** and **3t**. These molecules have better affinity than curcumin and galantamine which were evaluated as references. Therefore, **3f**, **3r** and **3t** were synthetized for their evaluation in vitro not only as inhibitors of AChE and Aβ_1-42_ aggregation but also as antioxidant agents. Finally, their cytotoxicity activity on PC12 cells was tested. The results showed that 3**r** inhibits AChE, avoids Aβ_1-42_ aggregation and exhibits antioxidant activity but shows cytotoxic effects. 

However, **3t** showed the best anti-Aβ_1-42_ aggregation, inhibited the AChE activity and was not cytotoxic in PC12 cells. Then, the chemical scaffolds of **3t** could be employed to design new molecules with multitarget activity. However, due to any of the selected compound being able to cross the BBB according to the predictors, these could be administrated using nanocarriers as liposomes or intranasal administration to reach the brain. 

## 2. Results

### 2.1. Interactions of Benzothiazole-Isothiourea Derivatives with AChE and Aβ_1-42_ by Docking Studies

The docking simulations of benzothiazole-isothiourea derivatives (Figure 1) were carried out on AChE and Aβ_1-42_ to search compounds with better affinity on these targets than their reported ligands. 

The affinity of the 22 benzothiazole-isothiourea derivatives against Aβ_1-42_ was evaluated in three conformations: α-helix, β-sheet and random coil (RC) (Appendix A). Figure 2 depicts the ΔG values (< ΔG values > affinity) for each of the compounds on Aβ_1-42_. It is important to obtain ΔG values and ligand binding modes on different Aβ_1-42_ conformations which are involved during its aggregation [24]. In the cell membrane, Aβ_1-42_ adopts an α-helix conformation; however, when it is delivered by the catalytic activity of gamma secretase (γ-secretase), it adopts structural changes to turn into β-sheet conformation passing for a random coil conformation [25]. Then, it is of utmost importance to identify compounds with more affinity for Aβ_1-42_ in α-helix conformation binding of the compound reaching to residues (E22 and D23) which are involved in the conformational changes. That binding mode could avoid the RC and β-sheet conformation inhibiting the Aβ_1-42_ aggregation [26]. There are other small molecules like curcumin which is a well-known herbal compound that has shown good binding with Aβ_1-42_ and prevents its aggregation [27,28]. Curcumin also decreases inflammation and cognitive deficits due to its anti-inflammatory and antioxidant properties [28]. The docking simulations showed that some benzothiazole-isothiourea derivatives have more affinity for Aβ_1-42_ than curcumin. The most promissory and interesting ligands were **3f**, **3q**, **3r**, **3t** and **3v** because they have more affinity for α-helix than the other Aβ_1-42_ conformations, these being **3q, 3f** and **3t** the best compounds according to the ΔG value. However, **3t** shows similar ΔG value on α-helix and β-sheet conformations (Figure 2a).

Regarding galantamine-Aβ_1-42_ complex, galantamine does not have better ΔG towards Aβ_1-42_ than **3f**, **3q, 3r** and **3t** compounds. 

In this work, curcumin shows a ΔG = −4.76 kcal/mol (Table 1); this result was comparable to other reports (−4 to −16 kcal/mol) [29], whereas **3f**, **3q, 3r** and **3t** show better affinity towards Aβ_1-42_ in α-helix conformation than curcumin (Table 1). 

For docking simulation on AChE, galantamine (AChEI) was used as a reference as it is already approved for AD treatment by the FDA. Galantamine increases the synaptic availability of acetylcholine (ACh) by inhibiting AChE competitively and reversibly. In addition, galantamine is also capable of inhibiting the Aβ_1-42_ aggregation [30,31]. Docking results of benzothiazole-isothiourea derivatives on AChE (Appendix A) showed that compounds **3f**, **3q**, **3r**, **3t** and **3u** (Figure 2B) had the best ΔG values. It is important to mention that galantamine exhibited a ΔG = −6.9 (Table 1) value comparable with previous in silico studies [32]. The best compounds (**3f**, **3t** and **3r**) were selected according to their favored ΔG values for AChE as well as for Aβ_1-42_ in the α- helix conformation (Figure 2). Despite that **3t** has similar ΔG for either Aβ_1-42_ in α- helix or β-sheet, it was selected as it has the best ΔG in AChE.

Docking simulations showed that curcumin binds in the opposite site (Figure 3a) compared to benzothiazole-isothiourea derivatives and galantamine on Aβ_1-42_, reaching a more negative density of Aβ_1-42_ due to their positive charges (Figure 3b) which are not present in curcumin (Figure 3d). However, all these ligands reach the structure region where the Aβ_1-42_ changes its α-helix conformation to acquire the β-sheet conformation (Figure 3c), which could avoid the Aβ_1-42_ aggregation. It is important to mention that the binding is due to benzothiazole-isothioureas interacts in α-helix Aβ_1-42_ with E22 and with K16 (Figure 3e), whereas curcumin makes a hydrogen bond with H14 and π-cation with K16 and hydrophobic interactions with L17 and A21.

Figure 3 shows the interaction of **3f** with E22, interacting with its sulfur (S) atom from benzothiazole; also, **3f** interacts with Q15 by making hydrogen bonds with amine groups and can form π-π interactions with F19, whereas its amantadine group makes interactions with H14 and V18 (Figure 3f). Regarding **3r**, the S atom from benzothiazole ring also makes interactions with E22 and the isothiourea group with D23 and reaches F19 and F20 residues (Figure 3g). On the other hand, **3t** makes similar interactions between the S atom and D23 and E22; additionally, its aromatic rings established π-π interaction with F19 and π-cation with H14 (Figure 3h).

It Is known that AChE has a catalytic anion site at the bottom of a narrow tunnel, lined mainly with aromatic residues, and is called a “gorge” which is the entrance to catalytic site/triad located at approximately 20 Å deep [33]. The catalytic triad is constituted by S203, H447 and E334. There are other protein regions which correct the orientation of the normal substrate (ionized acetylcholine) within the gorge, such as the oxyanion subsite or “oxyanion hole” (OAH) formed by A204, G121 and G122; the anionic subsite or “peripheral anion binding site” (PAS), which serves to orient the cationic part of acetylcholine, is located in the peripheral surface of the enzyme composed by W86, Y337, W286, Y72, D74 and Y341. Finally, there is a subsite or “acyl site” formed by W236, F338, F295, F297 and G122 where the acetyl group is binding [33]. All these sites regulate the catalysis of the enzyme. The docking simulations showed non-bonded interactions of galantamine towards AChE reaching the PAS site and the catalytic site [34]. In addition, **3f**, **3r** and **3t** were recognized in the same site as galantamine (Figure 4a).

The docking simulations showed that compounds reach the AChE gorge at the entrance of the catalytic site (Figure 4b). It is evident that **3f**, **3r** and **3t** established more chemical interactions (Figure 4) with the AChE active site than galantamine (Figure 4c), since **3f**, **3r** and **3t** are larger and have a larger variety of chemicals groups than galantamine. As well, the target ligands have sp3 bonds allowing free rotations between the benzothiazole rings. The previously mentioned structural features explain why **3f** (Figure 4d) reaches the PAS site by its benzothiazole ring interacting with Y337, D74, W286 and Y341. Additionally, **3f** reaches the acyl site by its amantadine group interacting with F338 and F297. 

On the other hand, **3r** reaches the PAS site, interacting with Y341, Y72 and D74 (Figure 4e). As well, **3r** reaches the acyl site interacting with F338 and F297. Additionally, there are two benzothiazole groups establishing interactions at the AChE gorge site. Finally, **3t** reaches the PAS interacting with D74, Y72, W286 and Y341 (Figure 4f) thanks to the presence of two benzothiazole rings leading to a stronger interaction compared with **3r** and **3t.** This suggests that the compounds could block the entrance of the AChE gorge and avoid the substrate to cross.

### 2.2. ADME, Toxicological and BBB Permeability Prediction

The selected compounds **3f**, **3r** and **3t** were submitted to the SwissADME server for physicochemical, lipophilicity, water solubility and pharmacokinetics properties prediction using its SMILE code (Appendix A). Out of the three compounds evaluated, only **3t** does not violate Lipinski’s rule of five for oral availability (Table 2). Furthermore, **3t** has a lipophilicity value (iLOGP) of 422; this is higher than **3r** and **3f** values (Appendix A). Thus, **3t** has low water solubility (Appendix A).

On the other hand, the results of the LD5O value and toxicity, carcinogenicity, immunotoxicity, mutagenicity and cytotoxicity of **3f**, **3r** and **3t** have been obtained through ProTox-II online (Table 2). The toxicity results show that the only compound classified as toxicity class 5 is **3t**. **3t** shows a LD50 of 4000 mg/k with cytotoxic activity with a probability of 0.52.

Finally, blood–brain barrier (BBB) permeability for all the selected compounds was obtained showing that any of the compounds can cross the BBB and that the gastrointestinal absorption was lower for **3r** and **3t** than for **3f** (Appendix A). 

### 2.3. Activity Assay of **3f**, **3r** and **3t** on AChE

The AChE activity was assessed with **3f**, **3r** and **3t**. It is important to mention that the activity was evaluated in presence of DMSO; this was employed to dissolve the compounds, despite reports indicating that DMSO could inhibit AChE [35]. As can be seen in the Appendix A the activity of AChE with and without the DMSO remain the same due to the amount of DMSO employed (0.4%). There are other organic solvents such as methanol that can be employed to test the AChE inhibitors [36]; however, these compounds showed the best solubility in DMSO. Additionally, DMSO was useful as the solubility of benzothiazole compounds is difficult up to 100 μM [37].

Therefore, all compounds were evaluated around 100 μM depending on its inhibitory concentration and its DMSO solubility. For example, **3f** was evaluated until 120 μM showing a Michaelis Menten behavior (Figure 5a) reducing the AChE activity. However, by applying the Lineweaver Burk equation, the graph shows (Figure 5b) a slight displacement of the line at 120 μM. Meanwhile, **3r** showed better inhibition parameters than **3f** according to the Michaelis Menten (Figure 5c) and Lineaweaver Burk (Figure 5d) graph at 100 μM. Regarding **3t**, the inhibitory effect on AChE was observed until 140 μM without affecting its solubility as occurred with the other compounds (Figure 5e,f). The inhibitory constant Ki for each compound was obtained as 0.1634 and 0.04929 for **3r** and **3t**, respectively; both being better than galantamine as was reported previously [36]. 

### 2.4. ThT Assay to Evaluated Aβ_1-42_ Aggregation with **3t**, **3f** and **3r** Compounds

The anti-Aβ_1-42_ aggregation effects of **3f**, **3t** and **3r** was evaluated using the ThT assay. First, the compounds were submitted to fluorescent experimental assays where any compound was fluorescent. Then, after 48 h incubation of Aβ_1-42_-compound complexes, there were no fluorescence effects observed. The emission at 480 nm was observed when ThT was added. Figure 6a depicts the fluorescence of the Aβ_1-42_ and Aβ_1-42-_**3r** complex at 50 μM over 48 h incubation; high fluorescence was observed even at a concentration of 100 μM of **3r**. Meanwhile, the Aβ_1-42-_**3t** complex at 100 and 50 μM showed lesser fluorescence than Aβ_1-42-_**3r** at 100 μM. The 100% of fluorescence intensity corresponds to free Aβ_1-42_ which decreased in presence of **3r** and **3t** at 100μM. Moreover, when **3t** was incubated at 50 μM, it showed significant difference versus Aβ_1-42_ alone (Figure 6b).

### 2.5. Antioxidant Activity of **3f**, **3r** and **3t** by DPPH and ABTS

The antioxidant activity by DPPH was evaluated for **3f**, **3r** and **3t** showing that only **3r** exhibited antioxidant activity. 5-asa was employed as a control showing 90% of DPPH reduction at 40 μM (Figure 7a). However, a higher concentration was necessary to observe a DPPH reduction employing compound **3r** which was able to reduce near to 60% of DPPH at 320 μM (Figure 7b). Moreover, the antioxidant activity by ABTS was assessed, observing that 5-asa was able to reduce near to 70% of ABTS radical at 160 μM (Figure 7c). Meantime, **3r** exhibited only 20% of the scavenging activity against ABTS radical (Figure 7d). 

### 2.6. Cytotoxic Activity of **3f**, **3r** and **3t** Compounds on PC12 Cell Line by MTT Assay

The cytotoxic activity of compounds was evaluated on the PC12 cell line using the MTT assay. The results show that **3r** is more cytotoxic than **3f** and **3t** on the PC12 cells (Figure 8 a). The cell viability for **3f** and **3t** was 100% at 100 μM for both; however, for **3f** the viability was 67.25% being significative at 100 μM. In addition, not only can the cell morphology be observed in Figure 8B but also the compounds precipitation at the end of incubation (37 °C, 48 h) is the degree of precipitation as follows **3r** > **3t** > **3f**.

## 3. Discussion

AD is one of the principal forms of dementia which will increase in the proceeding years. Unfortunately, there is only treatment for its symptoms. Different efforts have been made to find a multitarget compound to treat AD. AD is a multifactorial disease associated with multiple factors such as genetics, mitochondrial disfunction, oxidative stress, metal accumulation; enzymes such as MAO, BACE1, and AChE; proteins such as Tau and peptides such as amyloid peptide [38]. The actual drugs used to treat AD symptoms are AChE inhibitors. The design of multitarget drugs could take AChE as the principal target trying to reach additional biological targets, such as BACE1 involved in the Aβ_1-42_ production of MAO or GSK-3β and on the oxidative stress [10]. Galantamine is one of the principal drugs used for AD and is a competitive and reversible AChE inhibitor that interacts allosterically with nicotinic acetylcholine receptors. The pharmacological effects of galantamine include not only the improvement of the cognition function but also the facilitation of the activities of daily living in the short term (up to 6 months) in patients with mild to moderate AD. Thus, considering that AChE is one of the principal targets in AD and there are some benzothiazole tested as AChE inhibitors [39], in this work we have evaluated a set of benzothiazole-isothiourea derivatives as AChE inhibitors due to their pharmacological advantages. 

Therefore, taking into consideration that benzothiazole and isothioureas can act as AChE inhibitors, they were evaluated in silico with AChE; observing that these compounds interacted with amino acids residues from the PAS site, some of them showed better ΔG values than galantamine. Then, **3f, 3r** and **3t** were selected as the best AChE compounds. Derivatives **3r** and **3t** contain two benzothiazole rings which helped to interact in the gorge of AChE primarily in the peripheric site; additionally, the sulfur (S) atom played a key role to establish π-sulfur interactions with Y72 and Y341.

During in vitro assays, the principal problem with benzothiazole-isothiourea derivatives was their water solubility. It has been reported elsewhere that these compounds are soluble at 100 μM but at higher concentrations their solubility diminishes [37]. We observed that the solubility of **3r**, **3t** and 3**f** improved in DMSO compared to methanol. In addition, these compounds precipitated when evaluated in culture conditions; for other experimental conditions in which regular shaking and less time of incubation was employed, the compound precipitation was not observed. 

Furthermore, the in silico study of physicochemical properties, ADME and toxic and permeability properties was evaluated for **3f**, **3r** and 3**t**. The physicochemical properties of **3f**, **3r** and **3t** were examined in accordance with Lipinski’s rule of five where only the **3t** compound is considered a potential drug candidate because it satisfies the following properties: MW < 500 g/mol, Logp < 5, H-bonds donator < 5 and Hbond acceptor < 10 [40]. The toxicity prediction through the ProTox-II web server reveals that **3t** belongs to class V with LD50 of 4000 mg/kg and, despite it being predicted as cytotoxic in silico, in vitro analyses showed that it is not cytotoxic. However, any compounds can cross the blood–brain barrier permeability; therefore, the use of nanoparticles such as liposomes could be considered as an alternative for the administration of these compounds and, thus, more compounds can reach the central nervous system through intranasal administration.

Then, the benzothiazole-isothiourea was evaluated as an anti- Aβ_1-42_ aggregation knowing that Aβ_1-42_ is implicated in the amyloid cascade, which explains the formation of Aβ_1-42_ plaques during AD. This peptide is more hydrophobic than Aβ_1-40_, thus Aβ_1-42_ is more likely to form aggregates and is considered neurotoxic [41]. Therefore, molecules with benzothiazole group could be capable of making interactions with Aβ_1-42_, such as ThT [42]. In addition, new benzothiazole derivatives have recently been reported as inhibitors for Aβ_1-42_ aggregation and other enzymes for the treatment of AD [43]. The results obtained by docking simulations depict key interactions with the amino acids involved in the conformational change of Aβ_1-42_, such as D23 and E22, with the benzothiazole-isothiourea.

Therefore, the two aromatic rings present in the structure of **3r** and **3t** are important, which are also observed in the curcumin structure. As well, the presence of a linker between the aromatic rings from the benzothiazole groups played a key role to establish π–π interactions with F19 and F20, which explains their higher affinity according to their ΔG values. This finding correlates with findings reported by Reinke AA and Gestwicki JE in which it is described that the linker length between the two aromatic rings should be between 8 and 16 Å [44]. This feature is present in **3r** and **3t** compounds. Furthermore, the presence of a tertiary amine in the ring of **3t** can establish electrostatic interactions with E22 and D23 which could contribute to its favored ΔG values. In addition, the presence of aliphatic substituents in the aromatic rings helps to establish hydrophobic interactions with the methylene side chain of Lys16 and π-cation with the NH_3_ group. 

During the evaluation of Aβ_1-42_ fibril formation using the ThT assay in presence of **3r** and **3t**, it was showed that these ligands can avoid the Aβ_1-42_ fibril formation. The best compound was **3t** as was predicted by in silico studies; this could be explained by the presence of the two aromatic rings in the benzothiazole groups. Additionally, the linker between these aromatic rings contains a tertiary amine which is important to establish an interaction with E22. However, this does not apply to **3r**, in which case it has an aromatic ring, and did not establish interaction with E22. Therefore, the compound **3t** has more chemical groups that performed better Aβ_1-42_ anti-aggregation. 

In addition, the results from AChE inhibition showed that **3t** could be a better AChE inhibitor. However, it did not show antioxidant activity as **3r** did, in which the aromatic ring conjugates with the isothiourea group, in comparison with **3t** where this conjugation does not exist with the aromatic rings. 

Regarding cytotoxicity, it was observed that **3r** is more cytotoxic than **3t** in vitro, despite the in silico cytotoxicity prediction showing that **3t** could be more cytotoxic than **3r**. However, the LD50 was higher for **3t** than for **3r**. Therefore, **3t** could be a safe compound to be evaluated in vivo but also could be necessary to consider that in in silico prediction studies, **3t** could produce mutagenesis (0.51% probability); thus, studies about this should be conducted. 

Thus, **3t** could be evaluated in other targets, such as glycogen synthase kinase-3β (GSK-3β), as it has been reported that 1-aryl-3-benzylureas acts as GSK-3β inhibitors [45] and GSK-3β has also been used as a target for designing multitarget compounds for AD [46]. Due to this, this enzyme plays an important role during AD phosphorylating to Tau protein [47].

Therefore, the chemical structure of compound **3r** and **3t** results are interesting with respect to the pharmacophores to design a multitarget compound for AD targeting AChE, BACE1 and Aβ_1-42_ anti-aggregation, as it has been described that a linker between two aromatics rings containing a hydroxyethylene or hydroxyethylamine to form a hydrogen bond with the aspartic dyad in the catalytic site of BACE1 is necessary. In addition, the presence of two aromatic rings play a key role to establish a π-stacking interaction with clusters of aromatics residues present in the catalytic site and in the peripheral anionic site (PAS) of AChE [48].

Furthermore, as commented before, the anti-Aβ_1-42_ aggregation molecule should have aromatic rings separated by a linker; these chemical characteristics are present in **3r** and **3t** compounds (Figure 9).

Derivatives **3r** and **3t** have the advantage of having an aromatic ring or an imidazolidine as a linker, respectively. At each end of the linker there is a benzothiazole ring which helps not only to maintain more interactions in the PAS site of AChE but also the interaction with the aromatic residue of Aβ_1-42._ In addition, it is possible to observe the importance of the group at the end of the linker as for compound **3u** which the presence of a hydrocarbon chain in the linker lead to unfavorable inhibitory activities. 

## 4. Materials and Methods

### 4.1. In silico Evaluation

#### 4.1.1. Preparations and Optimization Ligand for Docking Studies 

The 2D structures of the ligands (22 benzothiazole-isothiourea) were drawn using ACD/ChemSketch 14.01 free software (Toronto, ON, Canada) [49], pre-optimized the structures once hydrogen atoms were added and converted to 3D to be saved in ∗.mol format. The matrix Z was generated for each ligand using the GaussView 5.0.9 program [50]. The structures were energetically minimized using a semi-empirical method (AM1). The 3D integrity of the molecules was verified after structure minimization and ∗.pdb file was generated. Finally, the structures were optimized using the Avogadro program [51] generating the *.pdb file to carried out docking simulation. 

#### 4.1.2. Protein Pre-Optimization for Docking Studies 

The 3D structure of AChE was obtained from Protein Data Bank [52]. PDB ID: 4PQE. The Aβ_1-42_ was considered in three conformations: α-helix, β-sheet and RC obtained from PDB 1Z0Q (alpha-helix), 2BEG (beta-sheet) and the RC conformation from the previous work of molecular dynamics simulation [53]. The proteins were prepared removing water molecules manually with a text editor. Then, the Gasteiger partial charges, polar hydrogen atoms and Kollman charges were added. Finally, the *.pdbqt. file was generated using AutoDock Tools 1.5.6 program (La Jolla, CA, USA) [49].

#### 4.1.3. Docking Studies 

For the docking studies, the proteins were rigid, whereas the ligands were flexible. The *.pdb, ∗.pdbqt, ∗.gpf and ∗.dpf files were created in AutoDock Tools. After the docking simulations, the protein-ligand interactions were evaluated using AutoDock Tools. The grid box was of 60 Å^3^ with a grid spacing of 0.375 Å^3^. For 4QPE the grid center was at X = −25.93, Y = 30.821, Z = −6.062. With this box, the residues H447, E334, S203, Y337 were included, whereas for 1Z0Q (alpha-helix) the grid center was X = 2.282, Y = 5.061, Z = −6.757; for 2BEG (beta-sheet) X = 2.937, Y = −4.619, Z = −1.241 and for 1Z0Q (RC) X = 9.387 Y = −4.642 Z = 1.805. The scoring sampling of docking study used the Lamarckian genetic algorithm as implemented in AutoDock Tools with an initial population of 100 individuals with 1 × 10^7^ evaluations. The ligand-protein complexes were analyzed to find the lowest free (ΔG) values by means of the AutoDock tools program to then describe the ligand-protein interactions. 

#### 4.1.4. Visualization of Protein-Ligand Interactions 

Pymol 2.5.2 software [54] and BIOVIA Discovery Studio Visualizer software [55] were used to visualize the ligand-protein interaction obtained from docking simulations. The interactions considered were less than a distance of 5 Å.

#### 4.1.5. ADME, Toxicological and BBB Permeability Prediction

The physicochemical properties of **3f**, **3r** and **3t** compounds were predicted via SwissADME [56]. The toxicological endpoints and the level of toxicity of **3f, 3r** and **3t** compounds studied were determined using ProTox-II server [57].

### 4.2. Reagents for Synthesis and In Vitro Evaluations

All reagents were purchased from Sigma Aldrich, Toluca, Mexico. 

#### 4.2.1. Synthesis of Benzothiazolilisothioureas Derivatives 

The synthesis of compounds **3f**, **3r** and **3t** was conducted as reported [58,59] with some modifications.

##### 1-Adamantan-1-yl-3-benzothiazol-2-yl-2-methyl-isothiourea (**3f**)

In a 100 mL flask, 1.0 g (3.94 mmol) of 2-dithiomethylcarboimidatebenzothiazole was dissolved with 20 mL of anhydrous ethanol. 3.94 mmol of adamanthylamine was added. The mixture was refluxed for 36 h. The solvent was reduced to 10 mL by evaporation and cooled to room temperature. After precipitation, the resulting solid was filtered and washed with a mixture of 1:1 water–ethanol. The compound **3f** was obtained as white crystalline solid, (0.844g), 60.0% yield, 95% purity, mp = 114–115 °C, ^1^H NMR [δ, ppm, CDCl_3_]: 12.51 (b, 1H, NH), 7.74 (d, 1H, H4), 7.71 (d, 1H, H7), 7.38 (t, 1H, H5), 7.26 (t, 1H, H6), 2.53 (s, 3H, SCH_3_), 7.44 (s, 4H, Ph). ^13^C NMR [δ, ppm, CDCl_3_]: 172.10 (C11), 164.56 (C2), 151.03 (C9), 136.29 (C15), 132.35 (C8), 127.46 (C16), 126.05 (C4), 123.80 (C5), 121.39 (C6), 120.82 (C7), 14.55 (SCH_3_), z/e (M + 1) = 358.14 (100%). NMR (Appendix A) and ESI-MS (Appendix A) 

##### 1-Benzothiazol-2-yl-3-[4-(3-benzothiazol-2-yl-2-methyl-isothioureido)-phenyl]-2-methyl-isothiourea (**3r**)

In a 100 mL flask, 1.0 g (3.94 mmol) of 2-dithiomethylcarboimidatebenzothiazole was dissolved with 20 mL of anhydrous ethanol. Then, 1.97 mmol of *p*-phenylenediamine was added. The mixture was refluxed for 24 h. The solvent was reduced to 10 mL by evaporation and cooled to room temperature. After precipitation, the resulting solid was filtered and washed with a mixture of 1:1 water–ethanol. The compound **3r** was obtained as yellow powder (0.678 g) 66.22% yield, 97% purity, mp = 232–233 °C. ^1^H NMR [δ, ppm, CDCl_3_]: 12.51 (b, 2H, NH), 7.74 (d, 2H, H4), 7.71 (d, 2H, H7), 7.44 (s, 4H16, Ph), 7.38 (t, 2H, H5), 7.26 (t, 2H, H6), 2.53 (s, 6H, SCH_3_). ^13^C NMR [δ, ppm, CDCl_3_]: 172.10 (2C11), 164.56 (2C2), 151.03 (2C9), 136.29 (2C15), 132.35 (2C8), 127.46 (2C16), 126.05 (2C4), 123.80 (2C5), 121.39 (2C6), 120.82 (2C7), 14.60 (2SCH_3_). νIR (cm^−1^, film): 1571 (vs, C11=N). Elemental analysis: Calculated: %C (55.35), %H (3.88), %N (16.14); Found: %C (55.43), %H (3.87), %N (16.08). z/e (M + 1) = 521.07 (7%). NMR (Appendix A) and ESI-MS (Appendix A) spectra.

##### 1-Benzothiazol-2-yl-3-{2-[2-(benzothiazol-2-ylimino)-imidazolidin-1-yl]-ethyl}-2-methyl-isothiourea (**3t**)

In a 100 mL flask, 1.0 g (3.94 mmol) of 2-dithiomethylcarboimidatebenzothiazole was dissolved with 20 mL of anhydrous ethanol. Then, 1.97 mmol of diethylenetriamine was added. The mixture was refluxed for 12 h. The solvent was reduced to 10 mL by evaporation and cooled to room temperature. After precipitation, the resulting solid was filtered and washed with a mixture of 1:1 water–ethanol. The compound **3t** was obtained as white crystalline solid (0.58 g), 63.04% yield, 96% purity. ^1^H NMR [δ, ppm, CDCl_3_]: 10.92 (b, 1H, NH), 8.86 (b, 1H, NH), 7.64 (d, 1H, H4), 7.62 (d, 1H, H4´), 7.60 (d, 1H, H7), 7.56 (d, 1H, H7´), 7.30 (t, 1H, H5), 7.25 (t, 1H, H5´), 7.16 (t, 1H, H6), 7.13 (t, 1H, H6´), 3.60–3.70 (m, 8H, H15, H15´, H17, H18), 2.54 (s, 3H, SCH_3_), ^13^C NMR [δ, ppm, CDCl_3_]: 174.60 (C11), 172.20 (C11´), 165.80 (C2´), 159.70 (C2), 152.16 (C9´), 151.33 (C9), 132.38 (C8, C8´), 125.54 (C4), 123.39 (C5), 122.39 (C6), 120.55 (C7), 121.2 (C17) 121.17 (C18), 125.80 (d, C4´), 119.49 (C7´), 47.04 (C15), 44.27 (C18), 42.54 (C17), 41.24 (C15´), 14.27 (SCH_3_), z/e (M + 1) = 468.11(100%). NMR (Appendix A) and ESI-MS (Appendix A) spectra.

### 4.3. In Vitro Assays 

#### 4.3.1. AChE Activity In Vitro Evaluation

An ACh curve was made from 0.8, 1.6, 3.2, 6.4, 12.8 and 16 μM and brought to 250 μL with phosphate buffer at pH 8.0. To quantify the ACh, 20 μL of an alkaline hydroxylamine solution (prepared at the time of use, mixing 1:1 volume of hydrochloride hydroxylamine 14% and NaOH 14%) was added and homogenized in vortex. After, 42 μL of the reaction mix was transferred to 96 well plate and 125 μL of FeCl_3_ (12.8 mg/mL with 12.8 % of HCl) was added and read in a Thermo Scientific, Multiskan Sky plate reader at 540 nm. The AChE kinetics were performed by adding 0.02 U AChE per reaction with the different ACh concentrations and, after the mix of reaction was incubated for 1 h at 37 °C and shook at 300 rpm, the ACh was quantified as mentioned before, adding the alkaline hydroxylamine solution and FeCl_3_. Finally, the compounds to be evaluated were added in the mix reaction with ACh, AChE at **3f:** 80, 100 and 120 μM, **3r**: 40, 60, 80 and 100 μM and **3t**: 80, 100 and 140 μM.

#### 4.3.2. Evaluation of Aβ_1-42_ Aggregation In Vitro by Thioflavin T (ThT) Assay 

Evaluation of ligands as Aβ_1-42_ fibril formation inhibitors was performed as follows: a solution of Aβ_1-42_ (Calbiochem, Cat. No. PP69) at 0.25 μg/μL in milliQ water was incubated with or without each compound (**3f, 3r** and **3t**) at 50 and 100 μM (DMSO < 0.1%) in a quartz cell at 37 °C. The mixture (300 μL) was constantly shaken over 48 h. Aliquots (150 µL) from this solution were taken at 48 h. Then, 25 µL of ThT at 3.0 μM was added and diluted to a final volume of 600 µL with miliQ water. The increase in ThT fluorescence was measured at λ emission = 480 nm and λ excitation = 445 nm [60]. Fluorescence emission was measured using an LS-55 Spectrofluorometer (PerkinElmer). All the experiments were performed using cells with a path-length of 0.5 cm, at room temperature.

#### 4.3.3. Antioxidant Evaluation 

##### 2,2-Diphenyl-1-picrylhydrazyl (DPPH) Assay

100 μL of DPPH (0.20 mM) in absolute methanol and 100 μL of each compound (0.32, 0.16, 0.08, 0.04, 0.02, 0.01 mM) dissolved in DMSO were poured into a 96-well plate in triplicate (A_1_). Another series with same concentration of compound in DMSO: methanol was used without DPPH (A_2_). In addition, into 3 wells were added 200 μl of DMSO: methanol (A_S_) and, finally, in other 3 wells were added 100 µL of DMSO: methanol and 100 μL of DPPH solution (A_DPPH_). All mixtures were incubated for 30 min at room temperature and protected from light. The absorbance was recorded at 517 nm in a transparent 96-well test microplate (Multiskan-EX Thermo Scientific, Thermo Fisher Scientific, Waltham, MA, USA). The results were expressed as percentage of DPPH radical reduced (antioxidant activity) for each concentration of the selected compounds. The percentage of the DPPH radical reduced was calculated using the following equation: [1 − ((A_1_ − A_2_)/(A_DPPH_ − A_S_))] × 100, where: A_1_ = Absorbance of the compound with DPPH, A_2_ = Absorbance of the compound plus DMSO: methanol, A_DPPH_ = Absorbance of DPPH (diluted 1:1 with DMSO: methanol) and A_S_ = Absorbance of DMSO: methanol [61].

##### 2,2-Azino-bis(3-ethylbenzothiazolin)-6-sulfonic Acid (ABTS) Assay

The ABTS radical cation (ABTS^+•^) was performed by mixing ABTS 7.00 mM with an aqueous solution of potassium persulfate 2.45 mM for 16 h at room temperature in the dark. After this time, a dilution (1:50 in DMSO: methanol) was made to allow ABTS to have an absorbance near to 0.7. In a 96-well plate, 100 μL of the corresponding compound (**3f, 3r** and **3t**) at 0.32, 0.16, 0.08, 0.04, 0.02 and 0.01 mM in DMSO was mixed with either 100 μL of diluted ABTS^+•^ solution (A_1_) or with DMSO: methanol (A_2_). 100 μL of diluted ABTS^+•^ radical was mix with 100 μL DMSO: methanol (A_ABTS_). The reaction was incubated for 30 min at room temperature protected from light. The absorbance was recorded at 734 nm in a transparent 96-well test microplate (Multiskan-EX Thermo Scientific) [61].

The antioxidant activity was calculated as the percentage of the ABTS ^+•^ reduced with the test compound using the following equation: [1 − ((A_1_ − A_2_)/(A_ABTS_ − A_S_))] × 100, where: A_1_ = Absorbance of the compound with ABTS; A_2_ = Absorbance of the compound with DMSO/methanol, A_ABTS_ = Absorbance of ABTS, A_S_ = Absorbance of DMSO/methanol. The reference compound used for DPPH and ABTS tests was 5-ASA. 

### 4.4. Cytotoxic Evaluation of Compounds on PC12 Cells

The PC12 cell line was grown in DMEM medium with fetal bovine serum 10% and 1X antibiotic-antifungal (penicillin G, sodium salt and 1% streptomycin sulfate) under 5% CO_2_ atmosphere at 37 °C. The cells were treated and visualized in a biosafety level 2 vertical laminar flow cabinet (NUAIRE A2 NU-543-400) and an inverted binocular microscope (MOTIC AE-20), respectively. To detach the cells, a PBS-trypsin solution was used. The cells were seeded in a 96 plate well with 1 × 10^4^ cells in each well. After 24 h the medium was replaced using different treatments: medium, medium + 0.02% of DMSO and medium with **3f**, **3r** and **3t,** at 6.25, 12, 25, 50, and 100 μM. Two independent experiments were performed with *n* = 24. For the viability test, 3-(4,5-dimethylthiazol-2-yl)-2,5-diphenyltetrazole (MTT) was employed as follows: the medium was replaced with 50 μL of a MTT solution (0.5 mg of MTT/mL of PBS). The 96 plate well was incubated for 4 h under a 5% CO_2_ atmosphere at 37 °C. Afterward, the MTT was removed and 50 μL of DMSO was added, to solubilize the formazan salts to be read in the spectrophotometer Multiskan Sky microplate (Thermo Fisher Scientific, Waltham, MA, USA) at 550 nm. 

### 4.5. Statistical Analysis 

The results are presented as the mean ± SE. All analyses were performed using the statistical program GraphPad Prism Version 5.00 software [62]. Analysis of variance (ANOVA) with the Dunnett´s Multiple Comparisons test for the groups and control experiments were used and significant statistical difference was considered with *p* < 0.05.

## 5. Conclusions

Combining two or more pharmacophoric moieties in one framework is a promising approach to obtain hybrid molecules that can be employed as multitarget compounds for AD treatment. Therefore, in this work, a benzothiazole group was combined with an isothiourea group to obtain 22 derivatives. Compound **3r** presented multitarget activity for AD-inhibiting AChE and Aβ_1-42_ aggregation and showed antioxidant activity, though it was cytotoxic on PC12 cells. On the other hand, compound **3t** showed a better performance against Aβ_1-42_ aggregation not only in in silico but also in vitro studies as an AChE inhibitor. In this case, no antioxidant activity or cytotoxic effects in vitro studies were observed despite in silico prediction suggesting otherwise with a 50% probability. 

In addition, the LD50 was higher for the **3t** compound, showing that the in vivo administration could be safe; therefore, **3t** could be employed either as a dual compound, evaluated in another target such as GSK-3β or be employed as a scaffold to design new molecules with multitarget activity.

Since in silico predictors showed that any of the compounds can cross the blood–brain barrier, in vivo administration could be performed either by using nanocarriers to arrive at the central nervous system or via the intranasal. Combining their transport within liposomes along with intranasal administration, an improvement in crossing the protein–lipid membrane can be observed specifically for 3**t** due to its higher lipophilicity in comparison with **3r** and **3f**.

## Figures and Tables

**Figure 1 ijms-23-12945-f001:**
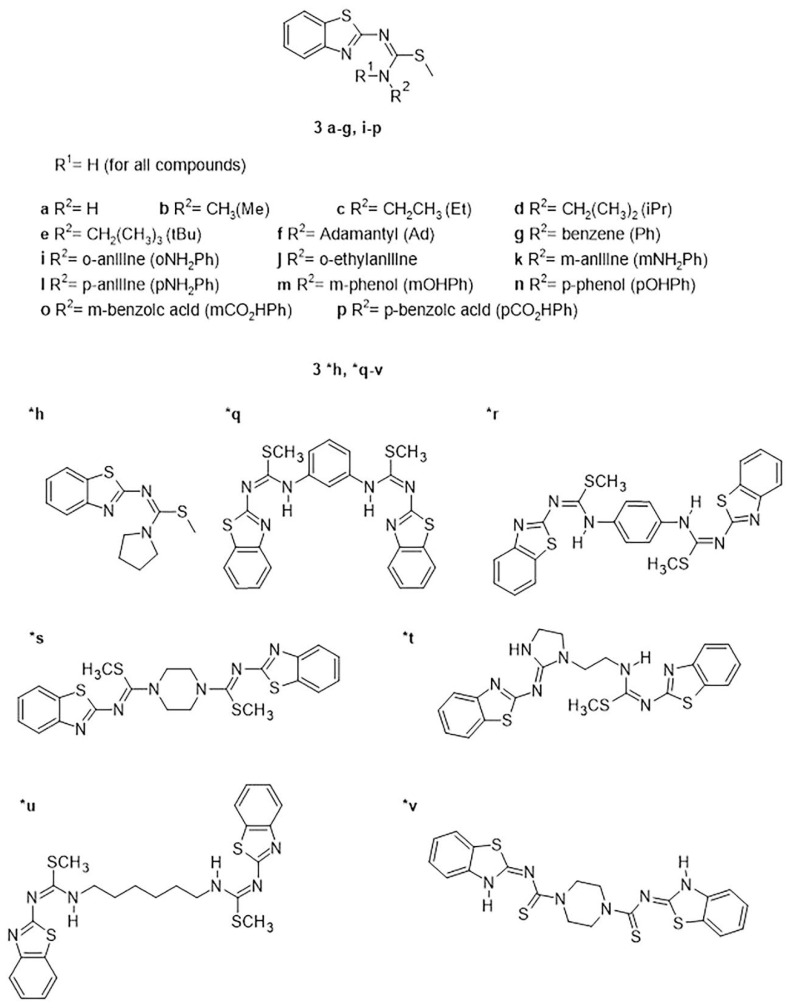
Chemical structure of the 22 benzothiazole-isothiourea derivatives. (*) Complete structures.

**Figure 2 ijms-23-12945-f002:**
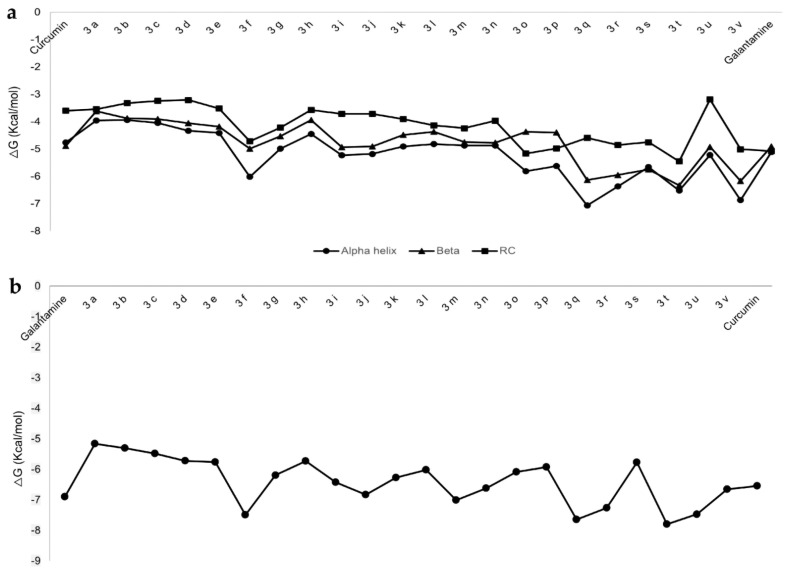
Free energy (ΔG; kcal/mol) values obtained by docking simulations. (**a**) Aβ_1-42_ in α-helix, β-sheet and RC conformations with the benzothiazole-isothiourea and its control compound curcumin. (**b**) AChE with the benzothiazole-isothiourea derivatives and its control compound galantamine.

**Figure 3 ijms-23-12945-f003:**
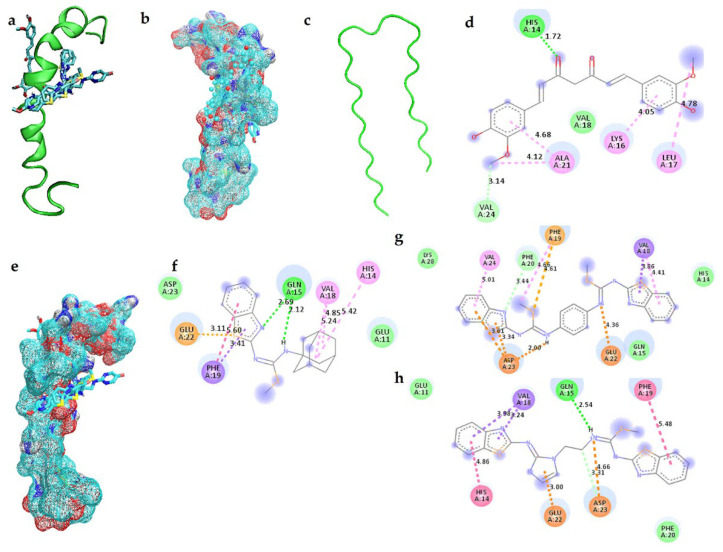
Non-bonded interactions obtained by docking simulations between Aβ_1-42_ in α-helix and the compounds. (**a**) Curcumin and **3f**, **3r** and **3t** compounds with the Aβ_1-42_ in α-helix conformation. (**b**) Curcumin recognized a protein surface with less negative density in the Aβ_1-42_ in α-helix conformation. (**c**) Aβ_1-42_ in β-sheet conformation. (**d**) Aβ_1-42_ in α-helix conformation and curcumin. (**e**) **3f**, **3r** and **3t** compounds recognized a protein surface with less positive density in the Aβ_1-42_ in α-helix conformation. Interactions of Aβ_1-42_ in α-helix conformation and: (**f**) **3f** compound; (**g**) **3r** compound; (**h**) **3t** compound.

**Figure 4 ijms-23-12945-f004:**
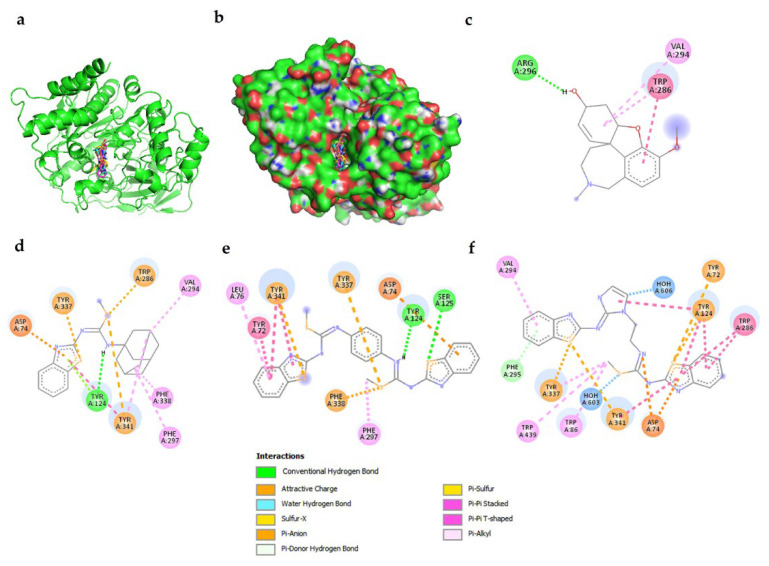
Non-bonded interactions obtained by docking simulations between AChE and the compounds. (**a**) Galantamine and **3f**, **3r**, **3t** compounds with AChE. (**b**) All compounds were recognized at PAS site near to the entrance of the gorge to the catalytic site. (**c**) AChE and galantamine. (**d**) AChE and **3f**. (**e**) AChE and **3r**. (**f**) AChE and **3t**.

**Figure 5 ijms-23-12945-f005:**
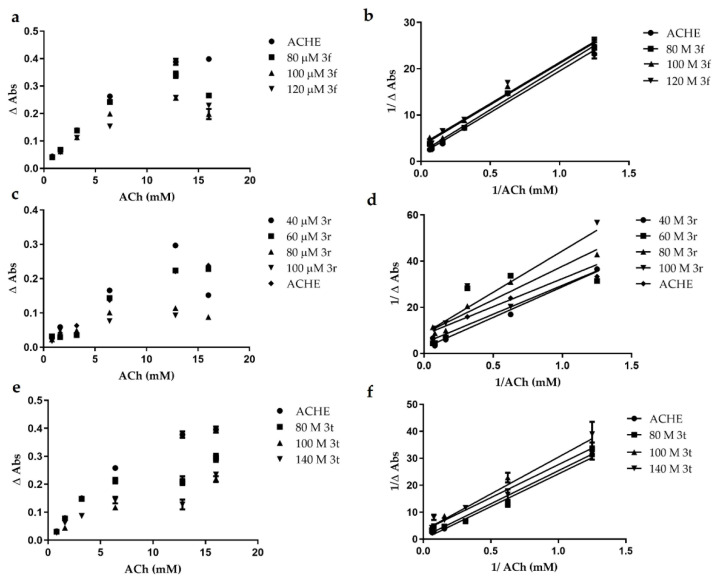
Enzymatic kinetic of AChE in presence of **3f**, **3r** and **3t**. Michaelis and Menten kinetic of compound (**a**) **3f;** (**c**) 3**r** and (**e**) **3t**; The Lineweaver Burk method for (**b**) **3f**; (**d**) **3r** and (**f**) **3t**.

**Figure 6 ijms-23-12945-f006:**
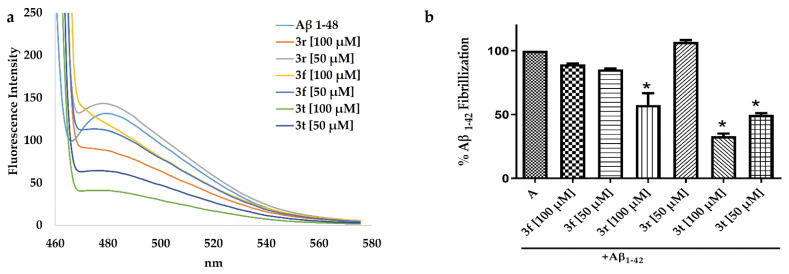
The anti-Aβ_1-42_ aggregation of **3f, 3t** and **3r** compounds. (**a**) Fluorescence intensity spectra of ThT and Aβ_1-42_ after 48 h of incubation at 37 °C. (**b**) Percentage of Aβ_1-42_ fibrillization taking 100% of the Aβ_1-42_ alone; the was percentage obtained after the incubation for 48 h with each compound. * Significant difference vs. Aβ_1-42_ alone (*p* < 0.05).

**Figure 7 ijms-23-12945-f007:**
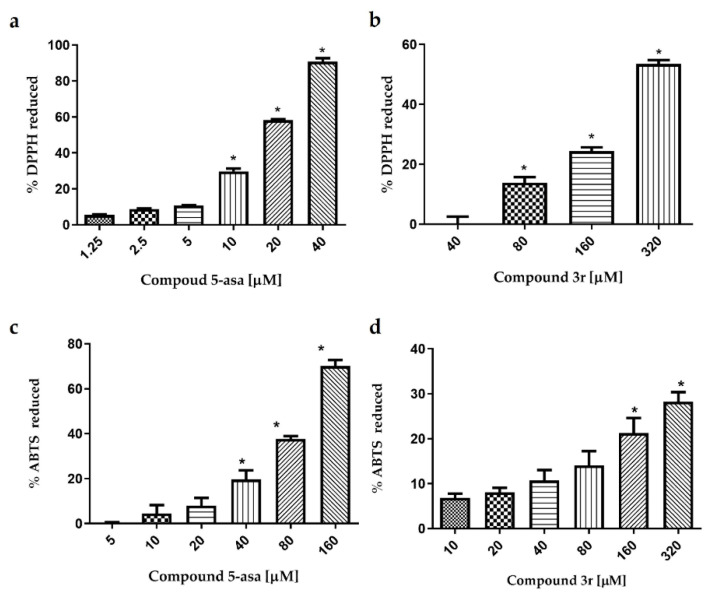
Antioxidant activity by DPPH and ABTS. (**a**) Percentage of DPPH reduced by 5-asa and; (**b**) **3r**. (**c**) Percentage of ABTS reduced by 5-asa and; (**d**) **3r**. The experiment was performed in triplicate in two independent experiments. * Indicate significant difference *p* < 0.05 between each concentration and the less concentration.

**Figure 8 ijms-23-12945-f008:**
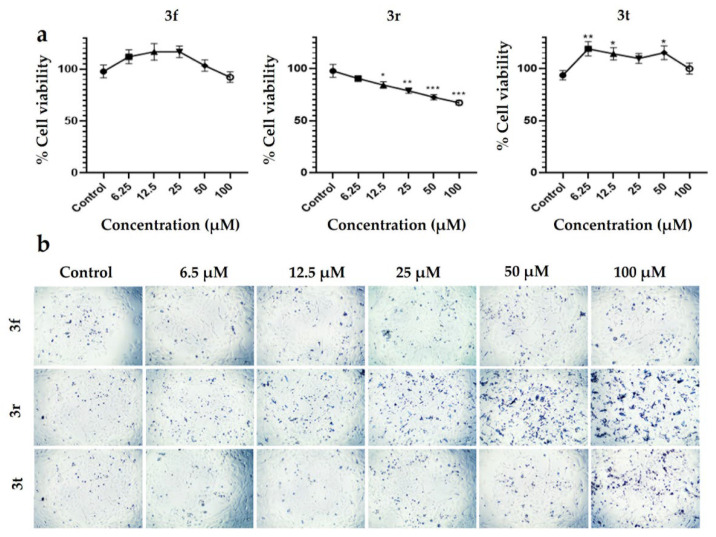
Cytotoxic effects of **3f**, **3r** and **3t** compounds at 48 h on PC12 cell line. (**a**) The cell viability by MTT assay for **3f**, **3r** and **3t** compounds on PC12 cells. (**b**) PC12 morphology posttreatments of **3f**, **3r** and **3t** compounds at 6.25, 12, 25, 50, and 100 μM after 48 h at 4× magnification. In the plots, each point represents mean with SEM. (* *p* < 0.05, ** *p* < 0.001, *** *p* < 0.0001), using Dunnett’s multiple comparisons test between different concentrations and the control.

**Figure 9 ijms-23-12945-f009:**
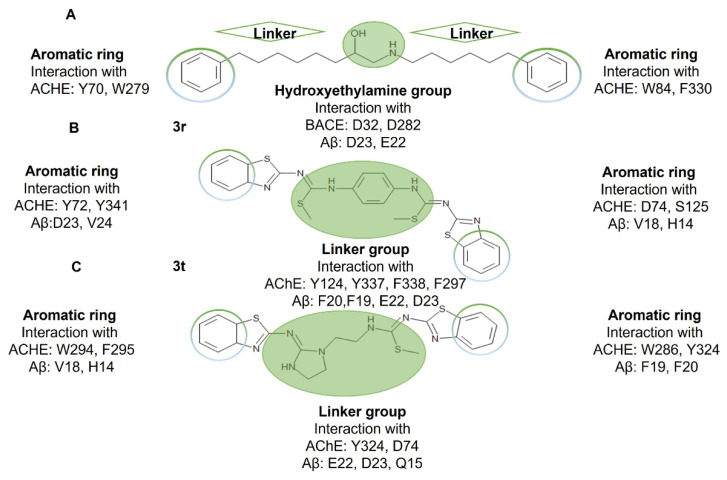
Compound **3r** and **3t** share chemical characteristics with the pharmacophore proposed to inhibit AChE, BACE1 and as anti-Aβ_1-42_ aggregation. (**A**) The chemical pharmacophore characteristics proposed to inhibit AChE, BACE1 and, in addition, the linker could create hydrogen bonds with D23 and E22 of Aβ_1-42_; (**B**) compound **3r;** and (**C**) **3t** had interactions with AChE and Aβ_1-42_ by its aromatic rings and the chemical groups located in the linker.

**Table 1 ijms-23-12945-t001:** ΔG values and amino acids of interaction obtained by docking simulations for the benzothiazole-isothiourea derivatives on Aβ_1-42_ in its α-helix conformation as well as on AChE.

Ligand	ΔG (kcal/mol)	Amino Acid Residues
Aβ_1-42_ in α-helix conformation		
Curcumin	−4.76	H13, H14, K16, L17, V18, A21, E22, V24, G25, S26
**3f**	−6.02	F20, F19, Q15, V12, H14, E11, D7, V18, E22, D23, N27
**3q**	−7.06	V12, E11, H14, Q15, V18, F19, F20, E22, D23, V24, N27, K28
**3r**	−6.37	N27, K28, D23, V24, A21, F20, E22, F19, Q15, V18, H14, E11, V12, Y10
**3t**	−6.52	Y10, E11, H14, Q15, V18, F19, F20, E22, D23, N27
AChE		
Galantamine	−6.9	Y341, S293, V294, F295, R296, F297
**3f**	−7.49	W86, D74, R296, F295, V294, Y341, F338
**3q**	−7.64	S293, V294, R296, F295, D74, F338, T83, N87, W86, G122, G121
**3r**	−7.26	G120, G121, R296, F295, V294, S293, Y341, F338, T83, D74, W86, N87
**3t**	−7.8	Y341, D74, V294, F338, T83, F295, G121, G122

**Table 2 ijms-23-12945-t002:** ADME, toxicological and permeability prediction of **3f**, **3r** and **3t** compounds.

Molecule	MW	#Heavy Atoms	#Aromatic Heavy Atoms	Fraction Csp3	#Rotatable Bonds	#H-bond Acceptors	#H-bond Donors	MR	TPSA	Lipinski #Violations
**3f**	357.54	24	9	0.58	4	2	1	105.76	90.82	1
**3r**	520.72	34	24	0.08	8	4	2	153.67	181.64	2
**3t**	467.63	31	18	0.24	7	4	2	141.39	159.58	0
Predicted Toxicity										
Molecule	Class	LD50: (mg/kg)	Carcinogenicity		Immunotoxicity		Mutagenicity		Cytotoxicity	
			Prediction	Probability	Prediction	Probability	Prediction	Probability	Prediction	Probability
**3f**	4	1000	Inactive	0.60	Inactive	0.99	Active	0.63	Inactive	0.77
**3r**	4	1000	Inactive	0.59	Inactive	0.98	Active	0.68	Inactive	0.77
**3t**	5	4000	Inactive	0.59	Inactive	0.94	Active	0.51	Active	0.52

## Data Availability

All results obtained from this studies were included in the article, and others are in the Appendix A, however if more information is needed it is available with the correspondence authors.

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
