# Peer review of "In Silico and In Vitro Studies of Benzothiazole-Isothioureas Derivatives as a Multitarget Compound for Alzheimer’s Disease"

_ijms, 2022, doi:10.3390/ijms232112945_

Round 1

Reviewer 1 Report

Comments are attached. 

Author Response

REVIEWER 1

Present work consists of the drug design of small molecules as multi-targeted ligands for the management of AD.

Major comments-

  1. Why Curcumin and galantamine were used as reference compounds. However, donepezil is marketed compound for AChE.

ANSWER: Donepezil, galantamine and rivastigmine are indicated in mild to moderate dementia due to Alzheimer disease, these drugs inhibit to AChE activity to avoid the Ach hydrolysis, however, we decide to use galantamine due it is an inhibitor of Ab1-42 oligomers, as well. (J Neurol Sci 2009, 15 280 (1-2):49-58. doi: 10.1016/j.jns.2009.01.024) (Mol Neurodegeneration 16, 2 (2021). https://doi.org/10.1186/s13024-021-00424-9)(Molecules 2020, 25, 5789; doi:10.3390/molecules25245789). Regarding to curcumin, it was used in this work due to the presence of some pharmacophore groups reported elsewhere to inhibit the amyloid beta aggregation and beside that is employed in several studies as anti-amyloid beta aggregation. (Biomolecules. 2020, doi: 10.3390/biom10091323. PMID: 32942739) (Journal of Biological Chemistry, https://doi.org/10.1074/jbc.M404751200)

Structures presented in fig 1 should be in same format like ACS or Willy etc.

ANSWER: Thank you for the suggestion, the figure 1 was corrected with the ACS format.

  1. Figure 6B, in ThT assay the error bars are generally higher. It is wearied to find such small error bars.

ANSWER: The fibrillization kinetics measured with ThT has oscillates values, which are molecular weight depending on being oligomers from low to high molecular weighs as occur naturally (doi.org/10.1080/13506129.2017.1304905). Then, we measure the ThT-Ab1-42  complex concentration in a quartz cuvette due to it is more reliable than microplate and a kinetic experiment was not done, we fix the ThT concentration and all samples were measured at the same time (http://dx.doi.org/10.1016/j.ab.2017.06.007). The experiments assays were carried out with the same batch of Ab1-42(calbiochem, merck), using one bottle for each experiment which were divided into 4 samples, (Ab1-42 + dissolvent, Ab1-42 + each compound in the dissolvent). All samples were incubated in a quartz cuvette and processed in the same conditions. The use of quartz cuvette and controlled procedure including agitation and temperature could explain the low error values as it has been reported that some experimental condition can affect the determination (http://dx.doi.org/10.1016/j.msec.2014). Furthermore, when the experiments are performed with a single concentration of compounds, similar standard errors can be obtained observing only those from the protein-protein interactions that can be affected in presence of compounds. (https://doi.org/10.1016/j.bmcl.2014.05.008).

  1. Figure 8a, compound 3f, as you are Increasing the concentration of compound 3f The percentage cell viability increases it may be the compound 3f is cell proliferation inducer.

ANSWER: The statistical analysis does not show significant difference among different concentrations employed despite that the at 12.5 mM show more cell viability than untreated cells.  Then, is not possible 3f increases the cell viability.

  1. Blood brain barrier permeability of the selected compounds should be established on the basis of in silico or in vitro experiments.

ANSWER: The blood brain barrier permeability of the target compounds has been screened by chemoinformatic analyses using public servers reported elsewhere, the methodology and the results were included in the new version of the manuscript.

Minor comments-

  1. English should be improved.

ANSWER: It has been revised carefully to improve the English language

  1. Relational of selecting 3a-g, i-p, q-w is not properly established. Only mentioning the pharmacological properties of benzothiazol and isothiourea groups and combining them followed by docking is not convincing drug design approach.

ANSWER: The multitarget compounds are usually designed through the framework combining two or more pharmacophoric moieties (hybrid molecule) enabling to reach two or more targets (Sampietro A. et al 2022). In addition, during the design of multitarget compounds to inhibit the Ab importance are given in the linked pharmacophore strategy (Pharmaceuticals 2022, 15,545.https://doi.org/10.3390/ph15050545). An example of this strategy is the ladostigil which is a combination of rivastigmine (AChE inhibitor) with rasagiline (MAO-B inhibitor). Another example is the combination of tacrine with curcumin among other examples with demonstrate advantages (MedChemComm, 2019,10, 2052. Doi.1039/c9md00337a)

Reviewer 2 Report

The manuscript entitled “In silico and in vitro studies of benzothiazole-isothioureas derivatives as a multitarget compound for the Alzheimer's disease” aims to design and evaluate 22 benzothiazole-isothiourea derivatives structurally guided by docking simulations selecting those capable of recognizing residues of the AChE active site and those involved in the Aβ1-42 aggregation with free radical scavenger activity (antioxidant activity) for their possible use as a multitarget treatment in AD. Besides, their cytotoxic activity was evaluated on PC12 cells. Considering the binding mode and the free energy (ΔG) values from in silico studies, the best compounds were selected. This topic is of high importance. The manuscript is overall prepared well. Still, I have some concerns. They are listed below.

Abstract:

-          “The Alzheimer disease (AD) is one of the principal dementias.” AD is not dementia. Please, rephrase.

-          Lines 21-24: the sentences are too long. Please, rephrase.

Introduction:

-          Lines 46-48: the sentences are too long. Please, rephrase.

-          Lines 86-91: the sentences are too long. Please, rephrase.

-          Figure 1: Structures should be visible better; they are too small.

The results are well presented.

The discussion is strong.

The methods are adequately described.

The Conclusion is too short. It should be written in more detail.

Author Response

REVIEWER 2

The manuscript entitled “In silico and in vitro studies of benzothiazole-isothioureas derivatives as a multitarget compound for the Alzheimer's disease” aims to design and evaluate 22 benzothiazole-isothiourea derivatives structurally guided by docking simulations selecting those capable of recognizing residues of the AChE active site and those involved in the Aβ1-42 aggregation with free radical scavenger activity (antioxidant activity) for their possible use as a multitarget treatment in AD. Besides, their cytotoxic activity was evaluated on PC12 cells. Considering the binding mode and the free energy (ΔG) values from in silico studies, the best compounds were selected. This topic is of high importance. The manuscript is overall prepared well. Still, I have some concerns. They are listed below. 

Abstract:

-          “The Alzheimer disease (AD) is one of the principal dementias.” AD is not dementia. Please, rephrase. 

ANSWER. The Alzheimer disease which has been defined as “Alzheimer’s disease (AD) (named after the German psychiatric Alois Alzheimer) is the most common type of dementia with slowly progressive neurodegenerative disease characterized by neuritic plaques and neurofibrillary tangles due to Ab1-42 production, oligomerization and accumulation principally in the brain (medial temporal lobe and neocortical structures). (Molecules 2020, 25, 5789; doi:10.3390/molecules25245789)

However, now we write as “Alzheimer disease is a progressive neurodegenerative disorder”

-          Lines 21-24: the sentences are too long. Please, rephrase. 

ANSWER: This observation was considered and done

Introduction:

-          Lines 46-48: the sentences are too long. Please, rephrase.

ANSWER: This observation was considered and done

-          Lines 86-91: the sentences are too long. Please, rephrase.

ANSWER: This observation was considered and done

-          Figure 1: Structures should be visible better; they are too small.

ANSWER: The figure 1 was corrected as was suggested also by the reviewer 1, using the ACS format

The results are well presented. 

The discussion is strong. 

The methods are adequately described. 

The Conclusion is too short. It should be written in more detail.

ANSWER: The conclusion was re-written considering this observation

Reviewer 3 Report

Dear authors,

I have read the manuscript entitled "In silico and in vitro studies of benzothiazole-isothioureas derivatives as a multitarget compound for the Alzheimer's disease”. The work focuses on the evaluation of a number of 22 benzothiazole-isothiurea (3-a-w) derivatives as inhibitors of acetylcholinesterase (AChE) and of the rest of the amino acids (Aβ1-42) in the beta-amyloid structure, through docking simulations. These derivatives could become multi-target compounds, useful in the therapy of Alzheimer's Dementia (AD).

Based on the results obtained from the in silico and in vitro studies, compound 3r could be used as a multitarget therapy because it was able to inhibit AChE, avoid Aβ1-42 aggregation, and exhibit antioxidant but cytotoxic activity. Compound 3t demonstrated activity against Aβ1-42 aggregation as well as AChE inhibitory activity, both in silico and in vitro, but without antioxidant activity and no cytotoxic effect. Compound 3t may be considered in the development of new molecules with multitarget activity for AD.

The paper complies with the requirements of the journal. The 9 figures and a table presented in the manuscript allow reading and a visualization of the described compounds. The bibliographic references used for documentation are in agreement with the chosen topic, some of them even being from recent studies. Although they were not mandatory, I appreciate that you also gave us some pertinent conclusions.

However, I have a few questions and suggestions:

1. For the selected compounds you demonstrated the relationship between chemical structure-activity, either on AChE, or on β-amyloid structure or other actions. A very important aspect is the calculation of ADME and toxicological properties in the site of the selected protein. Could the authors give us some information related to this topic?

2. Another model currently used for compounds obtained by virtual screening in is the (Q)SAR model. What do the authors think about this model?

3. For the 9 figures, I recommend a larger font size to increase the readability.

Author Response

REVIEWER 3

Dear authors,

I have read the manuscript entitled "In silico and in vitro studies of benzothiazole-isothioureas derivatives as a multitarget compound for the Alzheimer's disease”. The work focuses on the evaluation of a number of 22 benzothiazole-isothiurea (3-a-w) derivatives as inhibitors of acetylcholinesterase (AChE) and of the rest of the amino acids (Aβ1-42) in the beta-amyloid structure, through docking simulations. These derivatives could become multi-target compounds, useful in the therapy of Alzheimer's Dementia (AD).

Based on the results obtained from the in silico and in vitro studies, compound 3r could be used as a multitarget therapy because it was able to inhibit AChE, avoid Aβ1-42 aggregation, and exhibit antioxidant but cytotoxic activity. Compound 3t demonstrated activity against Aβ1-42 aggregation as well as AChE inhibitory activity, both in silico and in vitro, but without antioxidant activity and no cytotoxic effect. Compound 3t may be considered in the development of new molecules with multitarget activity for AD.

The paper complies with the requirements of the journal. The 9 figures and a table presented in the manuscript allow reading and a visualization of the described compounds. The bibliographic references used for documentation are in agreement with the chosen topic, some of them even being from recent studies. Although they were not mandatory, I appreciate that you also gave us some pertinent conclusions.

However, I have a few questions and suggestions:

  1. For the selected compounds you demonstrated the relationship between chemical structure-activity, either on AChE, or on β-amyloid structure or other actions. A very important aspect is the calculation of ADME and toxicological properties in the site of the selected protein. Could the authors give us some information related to this topic?

ANSWER: The ADME, toxicological and permeability properties for the selected compounds were calculated by chemoinformatic tools (doi.org/10.1038/srep42717) (doi.org/10.1093/nar/gky318) and now these are included and discussed into the new version of the manuscript.

  1. Another model currently used for compounds obtained by virtual screening in is the (Q)SAR model. What do the authors think about this model?

ANSWER: QSAR, SAR, and other studies lead to interesting chemical data from a set of reported ligands with the same chemical scaffold and similar pharmacological-biochemical responses. These can summit to quantum chemistry analyses to get structural, charges, polarity among others to identify structural and physicochemical properties associated to the biological responses, data which can be used for designing better new drugs. Then, in the future the combining pharmacophore of benzothiazole-isothiourea could be submitted to these studies to get more information for the target proposed or new targets related with AD.

For the 9 figures, I recommend a larger font size to increase the readability

ANSWER: Thank you for the observation the figure was corrected.

Round 2

Reviewer 1 Report

Not convinced by the authors on the provided explanations.  Currently, saw that NMR spectra are not integrated.   

Author Response

Please find enclosed the Response to reviewers of manuscript ijms-1937122 entitled “In silico and in vitro studies of benzothiazole-isothioureas derivatives as a multitarget compound for the Alzheimer's disease”. The reviewer 1 mentioned that our responses did not convince him, however, almost all our responses were supported by scientific articles.

Then, we only include the NMR integration into the supplementary material as suggested. Also, the English has been revised carefully.

Round 3

Reviewer 1 Report

Accepted